# A Planar-Symmetric SO(3) Representation for Learning Grasp Detection

**Tianyi Ko**[*]
Woven by Toyota, Inc.
`tianyi.ko@woven.toyota`

**Takuya Ikeda**[*]
Woven by Toyota, Inc.
`takuya.ikeda@woven.toyota`

**Hiroya Sato**
The University of Tokyo
`h-sato@jsk.imi.i.u-tokyo.ac.jp`

**Koichi Nishiwaki**
Woven by Toyota, Inc.
`koichi.nishiwaki@woven.toyota`

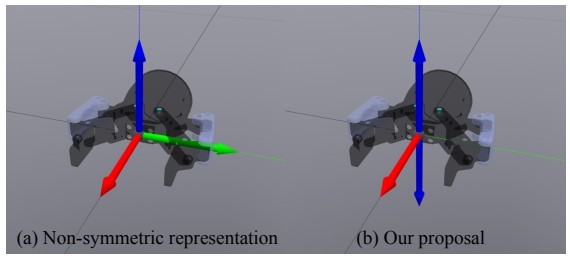 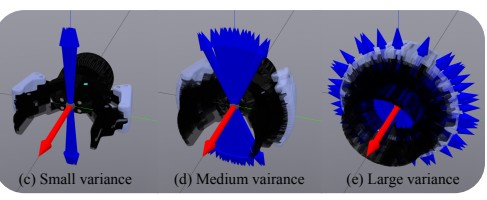

Figure 1: (a) For symmetric grippers, two distinct rotations (180°-flipped around the approach direction) representing the same grasp cause inconsistency and ambiguity. (b) We propose a novel planar-symmetric SO(3) representation that can express a pair of poses with a single parameter set. (c-e) It also provides deviation information, which is beneficial in the inference time.

**Abstract:** Planar-symmetric hands, such as parallel grippers, are widely adopted in both research and industrial fields. Their symmetry, however, introduces ambiguity and discontinuity in the SO(3) representation, which hinders both the training and inference of neural-network-based grasp detectors. We propose a novel SO(3) representation that can parametrize a pair of planar-symmetric poses with a single parameter set by leveraging the 2D Bingham distribution. We also detail a grasp detector based on our representation, which provides a more consistent rotation output. An intensive evaluation with multiple grippers and objects in both the simulation and the real world quantitatively shows our approach's contribution. A supplementary video is available at `https://youtu.be/24JZ9t7ZcI0`.

**Keywords:** Grasp Detection, Rotation Representation, Parallel Gripper

## 1 Introduction

Parallel grippers are commonly selected for grasping tasks due to their reliability, availability, and cost. For those grippers, flipping the gripper by 180° around the approaching direction results in the same grasp due to their planar symmetry, forming a bi-modal distribution in SO(3). While it is reported that having multiple feasible solutions introduces ambiguity [1] and a representation to explicitly manage such a multi-modal distribution is beneficial [2], many neural-network-based grasp detectors [3, 4, 5, 6, 7] directly regress the gripper rotation without enough consideration on this bi-modal ambiguity. In this paper, we propose a rotation representation that can represent the two feasible gripper rotations in a single parameter set, forming the problem into a uni-modal distribution.

---

[*]Equally contributed.

8th Conference on Robot Learning (CoRL 2024), Munich, Germany.

Bingham [8] proposed an antipodal symmetric distribution on the (hyper-)sphere. Peretroukhin et al. [9] and Sato et al. [10] discussed the application of 3D Bingham distribution for $\mathrm{SO}(3)$ representation by leveraging the fact that a pair of antipodal quaternions ($+\boldsymbol{q}$ and $-\boldsymbol{q}$) represent a single rotation in 3D. While [9, 10] handles *a single 3D rotation*, in this work, we propose a representation that can handle *a pair of 3D rotations* by leveraging 2D Bingham distribution to represent one of the basis vectors of a rotation matrix (See Fig. 1.)

Our approach is applicable to any grasp detector that *explicitly* regresses the gripper rotation. Without modifying the training data, it only requires (i) changing the network's rotation output channel to 9, (ii) replacing the rotation term of the loss function with our proposed one, (iii) additional eigenvector decomposition or sampling process for the inference time.

The contributions of this paper are summarized as follows:

1. We propose a novel $\mathrm{SO}(3)$ representation where one of the basis vectors of the rotation matrix is expressed by the 2D Bingham distribution so that a pair of two planar-symmetric poses are expressed by a single parameter set.

2. We propose a learned grasp detector with the planar-symmetric $\mathrm{SO}(3)$ representation. We qualitatively show its continuous properties and efficacy of the distribution information. We quantitatively show its superiority over the existing representation through simulation and real-robot experiments.

## 2 Related Works

### 2.1 Rotation Representations for Grasp Detectors

Morrison et al. [11] tackled 2D planar grasp generation and incorporated the symmetry of their two-fingered gripper by limiting the yaw angle representation in the range of $[0, \pi]$. Qin et al. [3] detected grasp poses in $\mathrm{SE}(3)$ by inferring the translational displacement and the rotation matrix [12] for each point on the object surface with a PointNet [13] backbone. Sundermeyer et al. [6] treated the point cloud as candidates of the gripper's contact points and estimated per-point rotation matrix. Breyer et al. [4] employed a 3D fully convolutional network to infer per-voxel rotation in quaternion form from a TSDF [14] input. Jiang et al. [5] followed [4] in the sense of rotation expression. Ko et al. [7] expressed the gripper rotation by the rotation matrix in their fully convolutional network. Our approach targets this line of work, in which the gripper rotation is explicitly learned by a neural network with a rotation loss function. In [3, 4], the planar-symmetry problem was already pointed out and tackled by selecting the smaller one of the two rotation losses for the ground truth rotation and the flipped one for the backpropagation. This paper quantitatively shows that our approach can better handle such problems.

There is another line of work for which our representation is not applicable. Zhao et al. [15] expressed the rotation by the direction the fingers move and the 1D angle around that direction. Wang et al. [16] first selected "view direction" that corresponded to the hand approach direction, then inferred the remaining 1D in-plane rotation against the cropped and reprojected point cloud, with both rotations discretized. Cai et al. [17] inferred five degrees of freedom (DoF) of the grasp pose in $\mathrm{SE}(3)$ by the location and the surface normal of the point cloud and discretized the remaining one DoF, which can handle multimodal grasp pose. Huang et al. [18] represented grasps as pairs of points on the point cloud and avoided direct rotation representation by analytically deriving it from the position and normal of the points. This work does not discuss which architecture is superior since it depends on the application, sensor, and hardware. Instead, we focus on providing a handy add-on to boost the performance of direct-rotation-regression-type grasp detectors.

### 2.2 Rotation Representations for Symmetric Objects

To our best knowledge, there are no symmetry-aware $\mathrm{SO}(3)$ representations for learning grasp detection. On the other hand, there are several such representations in the fields of object pose esti-

mation. Sato et al. [10] showed that the combination of the 3D Bingham distribution and a negative log-likelihood (NLL) loss can well represent the rotation of axial symmetry. However, a single 3D Bingham distribution can only handle a unimodal distribution over rotations. To handle a multimodal distribution, Riedel et al. [19] and Deng et al. [2] utilized the Bingham Mixture Model (BMM). Manhardt et al. [20] used multiple quaternions to represent multiple hypotheses of rotations that come from object symmetry. Zhang et al. [1] employed a continuous 6D rotation representation [12] and applied a diffusion model [21] to manage multiple feasible rotations. Although these approaches have the capability to handle the multimodality of rotations that come from discrete symmetries, they increase the number of hyperparameters or the complexity of the loss function compared to approaches with unimodal representations.

Since our target is the specific multimodal distribution of planar-symmetric grippers, which is a bimodal distribution, there can be a simpler and more efficient rotation representation without lack of capability. As such, our approach extends the work by Zhou et al. [12] by expressing one of its basis vectors with the 2D Bingham distribution to allow planar symmetry. Unlike the approaches based on a mixture distribution, the explicit constraint of antipodality to manage planner symmetry is unnecessary, thanks to the nature of 2D Bingham distribution.

## 3 Method

### 3.1 2D Bingham Distribution

The Bingham distribution [8] which has the property of antipodal symmetry is a probability distribution on the unit sphere $\mathbb{S}^{d-1} \subset \mathbb{R}^d$. We set $d = 3$ because we only consider $\mathbb{S}^2$ throughout this paper. The 2D Bingham distribution is parametrized by a $3 \times 3$ symmetric matrix. Letting $\mathrm{Sym}_n$ be the set of $n$-dimensional symmetric matrices, the parameter of the 2D Bingham distribution $A \in \mathrm{Sym}_3$ can be written with 6 parameters $\boldsymbol{a} \in \mathbb{R}^6$ using the function $\mathrm{triu} : \boldsymbol{a} \in \mathbb{R}^6 \rightarrow A \in \mathrm{Sym}_3$.

For every real symmetric matrix, we can choose an orthogonal matrix $D \in \mathbb{R}^{3\times3}$ and a real vector $\boldsymbol{\lambda} \in \mathbb{R}^3$ satisfying

$$A = D \operatorname{diag}(\boldsymbol{\lambda})D^\top, \tag{1}$$

where the operator $\operatorname{diag} : \mathbb{R}^3 \rightarrow \mathbb{R}^{3\times3}$ puts the elements of $\boldsymbol{\lambda}$ on the diagonal of the matrix. This decomposition can always be performed so that $\lambda_1 \leq \lambda_2 \leq \lambda_3$ is satisfied. Using this decomposition, the 2D *Bingham distribution* is defined as follows.

$$\mathfrak{B}(A)(\boldsymbol{v}) = \frac{1}{\mathcal{C}(\boldsymbol{\lambda})} \exp\left(\boldsymbol{v}^\top A \boldsymbol{v}\right), \tag{2}$$

where $\boldsymbol{v} \in \mathbb{S}^2$ and $A \in \mathrm{Sym}_3$. Here $\mathcal{C}(\boldsymbol{\lambda})$ is the *normalizing constant* of the Bingham distribution defined as below:

$$\mathcal{C}(\boldsymbol{\lambda}) = \int_{\boldsymbol{v}\in\mathbb{S}^2} \exp\left(\boldsymbol{v}^\top \operatorname{diag}(\boldsymbol{\lambda})\boldsymbol{v}\right) \mathrm{d}_{\mathbb{S}^2}(\boldsymbol{v}), \tag{3}$$

where $\mathrm{d}_{\mathbb{S}^2}(\cdot)$ is the uniform measure on the $\mathbb{S}^2$. Note that $\mathcal{C}(\boldsymbol{\lambda})$ depends only on $\boldsymbol{\lambda}$, the eigenvalues of $A$. Importantly, the below equation holds for any $c \in \mathbb{R}$,

$$\mathfrak{B}(D \operatorname{diag}(\boldsymbol{\lambda} + c)D^\top) = \mathfrak{B}(D \operatorname{diag}(\boldsymbol{\lambda})D^\top), \tag{4}$$

where $\boldsymbol{\lambda} + c = (\lambda_1 + c, \dots, \lambda_3 + c)$. Hence, the entries of $\boldsymbol{\lambda}$ can be shifted to satisfy the below.

$$\lambda_1 \leq \lambda_2 \leq \lambda_3 = 0 \tag{5}$$

It follows directly from the Rayleigh's quotient formula [22],

$$\arg\max_{\boldsymbol{v}\in\mathbb{S}^2} \boldsymbol{v}^\top (D \operatorname{diag}(\boldsymbol{\lambda})D^\top)\boldsymbol{v} = \boldsymbol{v}_{\lambda_3}, \tag{6}$$

where $\boldsymbol{v}_{\lambda_3}$ is a column vector of $D$ corresponding to the maximum entry of $\boldsymbol{\lambda}$. The mode vector $\boldsymbol{v}_{\lambda_3}$ coincides with the right-most column vector of $D$ when $\boldsymbol{\lambda}$ is sorted as Eq. 5. In the following parts, it is assumed that the condition Eq. 5 is always met.

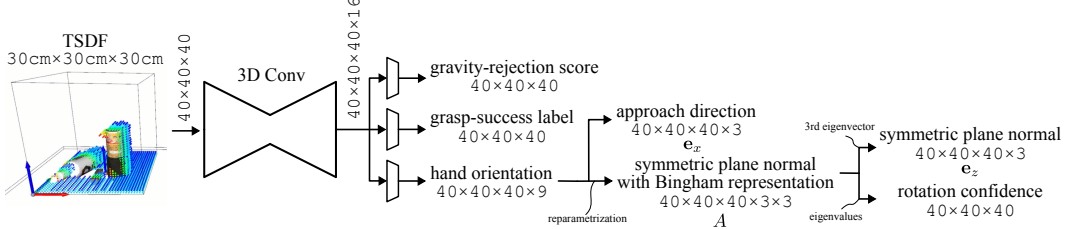

Figure 2: Architecture of our grasp detection network with the hand symmetric plane's normal vector $\boldsymbol{e}_z$ expressed in the 2D Bignham representation. We can either sample $\boldsymbol{e}_z$ from the Bingham distribution $\mathfrak{B}(A)$ or directly perform eigenvalue decomposition of $A$ and take the eigenvector with the largest eigenvalue as $\boldsymbol{e}_z$ while taking the difference of the three eigenvalues as the confidence.

## 3.2 A Planar-Symmetric SO(3) Representation and Its Loss Function

We propose a novel planar-symmetric SO(3) representation that can express a pair of poses with a single parameter set. Our proposal is a 9-parameter representation as follows.

$$[\boldsymbol{a}^\top, \boldsymbol{x}^\top]^\top \in \mathbb{R}^9, \quad \text{where} \quad \boldsymbol{a} \in \mathbb{R}^6, \boldsymbol{x} \in \mathbb{R}^3. \tag{7}$$

The 6-parameter vector $\boldsymbol{a}$ is the parameter of 2D Bingham distribution, that are described in Section 3.1. We make the mode antipodal-vector of the 2D Bingham distribution orthogonal to the symmetric plane. The 3 parameters $\boldsymbol{x}$ are a unit 3D vector that is parallel to the symmetric plane.

To train a neural network with our planar-symmetric SO(3) representation, we first sample a pose from a training dataset and express it as a rotation matrix $[\boldsymbol{e}_x \, \boldsymbol{e}_y \, \boldsymbol{e}_z] \in \mathbb{R}^{3\times3}$ where, using $\boldsymbol{x}$ and $\boldsymbol{a}$ defined in Eq. 7, $\boldsymbol{e}_x$ corresponds to $\boldsymbol{x}$ and $\boldsymbol{e}_z$ is a mode vector of $\mathfrak{B}(\mathrm{triu}(\boldsymbol{a}))$. The loss function is then expressed as

$$\mathcal{L}_{\cos}(\boldsymbol{x}, \boldsymbol{e}_x) + \mathcal{L}_{\mathrm{BNLL}}(\mathrm{triu}(\boldsymbol{a}), \boldsymbol{e}_z) \tag{8}$$

where $\mathcal{L}_{\cos}(\boldsymbol{a}, \boldsymbol{b}) = \boldsymbol{a}^\top \boldsymbol{b}/(\|\boldsymbol{a}\|\|\boldsymbol{b}\|)$ denotes the cosine similarity loss and $\mathcal{L}_{\mathrm{BNLL}}$ denotes the 2D Bingham negative log-likelihood loss (see Appx. B for more details). Note that while $\mathcal{L}_{\cos}$ has the same input size for $\boldsymbol{x}$ and $\boldsymbol{e}_x$, $A$ ($= \mathrm{triu}(\boldsymbol{a})$) and $\boldsymbol{e}_z$ have a different sizes for $\mathcal{L}_{\mathrm{BNLL}}$ because we implicitly represent $\boldsymbol{e}_z$ with the symmetric matrix $A$. In addition, since $\mathcal{L}_{\mathrm{BNLL}}(A, \boldsymbol{e}_z) = \mathcal{L}_{\mathrm{BNLL}}(A, -\boldsymbol{e}_z)$, we only need to sample one pose per each symmetric pair from the training data. In Eq. 8, we add the two losses without scaling. Even though cosine and log have different natures, we empirically observed that $\mathcal{L}_{\mathrm{BNLL}}$ typically converges from a range of 2-3 after the first epoch to a range of $\pm0.5$, which is close to that of $\mathcal{L}_{\cos}$. Our quantitative evaluation in Sec. 4 shows that this simple addition works well for the grasp detection.

## 3.3 Grasp Detector with Planar-Symmetric SO(3) Representation

In this section, we introduce our rotation representation to [7]. Figure 2 illustrates its network architecture. The network takes a $40\times40\times40$ TSDF [14] volume as input and employs a 3D fully-convolutional encoder-decoder feature extractor to acquire a $40\times40\times40\times16$ channel feature volume. The feature volume is further processed by three distinct fully convolutional heads. A gravity-rejection score head estimates, for each voxel, the magnitude of disturbance in the gravity direction that the grasp can resist if the TCP (tool center point) is located at that voxel. The grasp-success classification head classifies whether the grasp is feasible. Lastly, the grasp-rotation head estimates the per-voxel gripper rotation. More training details are in Appx. C.

In the original work [7], the grasp rotation head outputs 6 channels, which is a concatenation of $\boldsymbol{e}_x$ and $\boldsymbol{e}_z$ (red and blue vector in Fig. 1 (a), respectively) of the rotation matrix. To introduce the planar-symmetric SO(3) representation, we extend the output channels to 9 to parameterize Eq. 7. At the training time, we feed the network output $\boldsymbol{x}$ and $\boldsymbol{a}$ to Eq. 8. At the inference time, we perform an eigenvalue decomposition on $A$ to acquire its eigenvalues $\lambda_i$ and their corresponding eigenvectors $\boldsymbol{v}_{\lambda_i}$.

The implementation of $\mathcal{L}_{\text{BNLL}}$ is based on [23], with a minor modification to support 2D Bingham distribution. While the original code takes a few days to train our network, re-writing the NumPy-based operations with CuPy [24] makes the training time comparable to other rotation representations. With an Nvidia V100 GPU, four training runs took an average of 190 minutes, 90 minutes for the minimum case and 350 minutes for the maximum case. The rotation matrix versions took 200 minutes in average, and the quaternion version took 170 minutes. Considering the variance, the training cost overhead of our representation can be regarded as negligible. The overhead on the memory footprint is also small since our representation only changes the last layers of the network.

There are three ways for reconstructing the hand rotation during the inference time. The simplest one is to directly use the primary eigenvector $v_{\lambda_3}$ as $e_z$. We analyze its effect in Section 4.1. In order to make use of the information on the uncertainty, we can introduce a confidence threshold derived from $\lambda$. A mask to reject voxels with low confidence can be added to the output of the grasp-success classification head. We quantitatively evaluate this second approach in Section 4.3. The third approach is to sample $e_z$ from $\mathfrak{B}(A)$ based on the method in [25]. This solves a major limitation of the original works: each voxel can only represent one grasp pose. We evaluate this case in Section 4.2. In this case, in exchange for lower sample efficiency, we can get a diverse set of grasp poses, which is beneficial for considering additional constraints such as placement and collision avoidance.

In any case, there is no guarantee that $e_z$ is orthogonal to the network output $e_x$. Following [12], we perform a Gram-Schmidt orthogonization to reconstruct a valid rotation matrix. [7] reported that keeping $e_x$ and modifying $e_z$ provides a superior performance since $e_x$ typically coincides with the normal of the object's surface and is easier for the network to learn. We follow this approach by first normalizing $e_x$ then modifying $e_z$ as $e_z - (e_x \cdot e_z)e_x$.

## 4 Experiment and Evaluation

This section evaluates our approach with the vanilla version [7] as our baseline, which uses the rotation matrix representation proposed by [12]. For a more comprehensive comparison, we also trained a quaternion version, similarly to [4]. To train the baselines, we followed [3, 4] where we computed the rotation losses for both non-flipped and 180°-flipped rotation, then selected the smaller one for the backpropagation.

### 4.1 Analysis of Rotation Continuity

This section compares the continuity of the rotation volume output. A 60×60×200mm-sized box-shaped object was placed at the workspace origin, and a single depth image was captured from $[0.5, 0, 0.5]$ m. Figure 3 shows a comparison of our approach and the baseline rotation matrix version by plotting the gravity-rejection score field and gripper rotation field at $z = 57$ mm. In this case, the basis vector $e_z$ of the hand's rotation matrix (blue vector in Fig. 1 (a)) must be aligned with the y-axis to grasp the thin and long object.

The rotation field for the baseline is not consistent: some voxels have $+y$ output, and others have $-y$. This is natural because the network is trained to regress either of them, and both of them are the correct answer. The problem is that some voxels (surrounded by the red square) have an intermediate value of the two correct modes. If those voxels are selected for the grasp execution, the grasp will fail. In the case of our proposal, on the other hand, the vector field is consistent and smooth. Notice that for the proposed representation $e_z$ and $-e_z$ is expressed by the same parameter set (see Fig. 1 (b)); therefore, we plot both.

### 4.2 Analysis of Rotation Distribution

In order to analyze the distribution of the grasp rotation field, we placed a 60 mm tall and 70 mm diameter flat cylinder next to the thin and long box and acquired the grasp orientation field in the same process as Section 4.1. This time, we sampled 30 $e_z$ from $\mathfrak{B}(A)$ and overlaid them in Fig. 4.

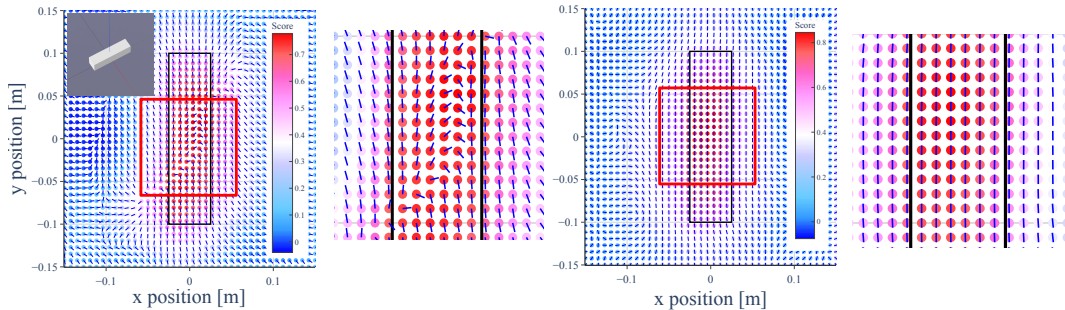

Figure 3: Cross-section of the network output grasp score field and rotation field at $z = 57$ mm plane when a long box is aligned with the workspace origin. Left: baseline. Right: ours.

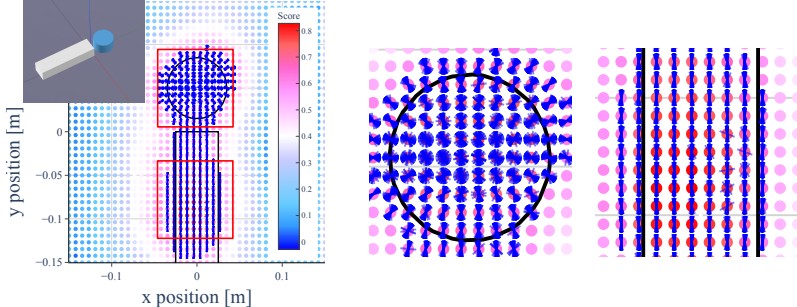

Figure 4: Same plot as Fig. 3 but a flat cylinder is placed next to the box. This time we sample 30 $e_z$ from $\mathfrak{B}(A)$. The uncertainty on the box is small because there are no other possible choices to grasp the long box, which corresponds to the case in Fig. 1(c). The distribution on the center of the cylinder is close to a uniform one because any downward grasp on the region is affordable. This corresponds to the case of Fig. 1 (e).

For the voxels on the thin and long box, the uncertainty is small and $e_z$ converges to a single grasp rotation. This is reasonable because the grasp will only succeed when $e_z$ is aligned with the y-axis, otherwise the finger will collide with the box. This matches the case illustrated in Fig. 1 (c) well. On the other hand, the uncertainty is large for the voxels on the cylinder. The uncertainty is especially large when it is aligned with the center of the cylinder, allowing arbitrary hand rotation around the $z$-axis. This corresponds to the case illustrated in Fig. 1 (e). For those near the edge of the cylinder, the uncertainty takes an intermediate value. While this may be reasonable, such inaccurate poses may cause grasp failures. As the uncertainty can be analytically acquired by the ratio of the eigenvalues, a proper threshold to reject those poses can be effective. Such cases are illustrated in Fig. 1 (d).

## 4.3 Quantitative Evaluation with a Physics Simulation

We performed experiments in simulation to compare our proposal with our baseline quantitatively. We followed the experiment protocol of existing works [4, 3, 15, 17, 18, 5, 7]: multiple objects are randomly dropped to a scene, and the robot declutters them into a bin. The performance is measured by two metrics: success ratio (SR), which represents the number of successful grasps divided by the number of total trials, and clear ratio (CR), which represents the number of successful grasps divided by the total number of objects. We used Drake [26] with a hydroelastic contact model [27] as the simulator. We simulated the whole declutter process, including the pre-contact motion, the interaction between the object and the hand's passive parallel link mechanism, and the lift motion until the object is dropped into a container placed near the workspace. Before each grasp, we rendered a single depth image to mimic the partial observation in the real world.

We trained the networks with two grippers: (i) Robotiq 2F-85 gripper with its passive compliance mode activated as an example of an underactuated gripper, and (ii) Franka-Emika's gripper, as an

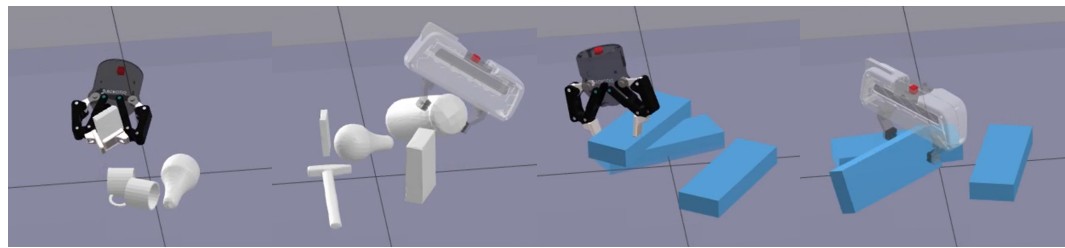

Figure 5: Capture the simulation experiment with two kinds of grippers and objects.

example of a rigid parallel-jaw gripper. We created two evaluation datasets: (i) one consisting of a subset of meshes provided by Breyer et al. [4] to evaluate the grasp performance on household objects, (ii) one with large flat boxes that require a highly accurate grasp pose due to small clearance (see Appx. D for more details).

Figure 5 shows simulation captures, and the supplementary video contains more examples. Table 1 summarizes the results. "2D Bingham" represents the case where we used the primary eigenvector $v_{\lambda_3}$ as $e_z$ of the rotation matrix; "2D Bingham w/ Conf. Tresh." represents the case with an identical setup to "2D Bingham," except that the voxels where $(\lambda_3 - \lambda_2) + (\lambda_3 - \lambda_1) < 15$ were masked out as an invalid grasp. Qualitatively, we observed two major failure modes. One was an incorrect approach direction corresponding to an inaccurate $e_x$. We could see some improvement in the rotation matrix version over the quaternion version but could not see the advantage of our approach. This is natural because our approach does not contribute to this. The other failure mode was an inaccurate yaw angle corresponding to $e_z$, frequently occurring when the object was laid on the surface and required a top-down grasp. We observed a clear improvement in our approach for this failure mode. We also saw that despite the underactuated gripper being effective for power grasping, its compliance is fragile against collision between the fingertip and the object during the approach. The result that our method had the highest effect for the combination of Robotiq gripper and large-flat boxes, quantitatively suggests that it improves the hand's yaw accuracy. As a reference, we also tried the 3D Bingham representation proposed by [10]. Surprisingly, it underperformed all other representations, indicating that naively adopting the Bingham distribution is not beneficial.

Table 1: Success ratio (SR) and clear ratio (CR) in simulation

|  |  | Robotiq 2F-85 Hand | | Franka-Emika Hand | |
|---|---|---|---|---|---|
|  |  | VGN Dataset | Large Box | VGN Dataset | Large Box |
| 2D Bingham (Ours) w/ | SR [%] | **79.0** | **62.9** | 63.1 | **67.7** |
| Conf. Thresh. | CR [%] | **76.6** | **70.6** | 64.9 | **80.6** |
| 2D Bingham (Ours) | SR [%] | 73.7 | 54.3 | **63.6** | 66.2 |
|  | CR [%] | 74.9 | 59.9 | **66.6** | 79.4 |
| Rotation Matrix [12, 7] | SR [%] | 65.5 | 46.1 | 58.7 | 48.7 |
|  | CR [%] | 70.2 | 52.0 | 63.8 | 57.5 |
| Quaternion [4] | SR [%] | 58.3 | 22.2 | 47.8 | 38.5 |
|  | CR [%] | 62.4 | 24.6 | 48.2 | 48.4 |
| 3D Bingham [10] | SR [%] | 45.6 | 16.4 | 39.1 | 20.5 |
|  | CR [%] | 47.4 | 17.5 | 39.8 | 23.4 |

## 4.4 Real-Robot Evaluation

To validate our approach in the physical world, we performed a real-robot evaluation with a Franka-Emika Panda robot, a Robotiq gripper with custom fingertips, and a wrist-mounted ZED X Mini stereo camera. We selected nine kinds of household objects from YCB [28] dataset. We first created a random scene in the simulator to ensure a random yet consistent arrangement of objects for all the test cases. Before each experiment, we manually aligned the objects in the real world to match

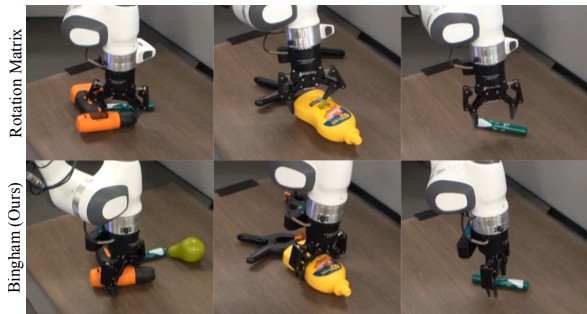

Figure 6: Examples of failure modes that happen only for the baseline but not for ours.

| | | |
|---|---|---|
| Bingham (Ours) w/ | SR | **70.0%** |
| Conf. Thresh. | $\overline{\text{CR}}$ | **74.5%** |
| Rotation Matrix | SR | 53.6% |
| | $\overline{\text{CR}}$ | 58.8% |
| Quaternion | SR | 55.9% |
| | $\overline{\text{CR}}$ | 64.7% |

Table 2: Success ratio (SR) and clear ratio (CR) against rotation expressions in real-robot evaluation

the scene created in the simulation. For each grasp, the robot first moved the wrist to a predefined pose and captured a single depth image. We then performed the inference and sorted the voxels in the gravity-rejection-score order, searching for the first applicable robot trajectory that fulfills: (i) the gripper linearly moves down in the -z direction 30 cm to the pre-grasp pose, (ii) the gripper moves forward for 10 cm to grasp the object, (iii) the gripper is linearly lifted by 30 cm. In order to avoid collision with the environment, we first used the camera to scan the table and used a planar patch detection [29] provided by Open3D [30] to acquire a set of static collision shapes. We did not explicitly avoid the collision between the gripper and objects other than the object to be grasped since colliding grasps are expected to be rejected during network inference [4, 7].

Table 2 summarizes the result, and a full video of the experiment is included in the supplementary material. These results prove that our representation also outperformed the baseline in the real world. Qualitatively, our approach worked better in three typical top-down grasp cases: (i) a T-shaped object, such as the electric drill; (ii) a large and flat object with limited position error tolerance, such as the laid-down mustard bottle; (iii) a small and thin object, such as the whiteboard marker. Similarly to the case in simulation, the baselines occasionally provided a clearly wrong yaw rotation, which could not be seen for our representation. Figure 6 shows some typical cases. Appendix E provides more details on target objects and their major failure modes.

## 5 Limitation and Scope of This Work

Our work only applies to planar-symmetric grippers since it relies on the nature of the Bingham distribution to represent a pair of antipodal vectors. Thus, it is not applicable to the works using three-fingered grippers or anthropomorphic hands. Another limitation is that our work applies only to grasp detectors that explicitly regress the gripper rotation. For example, EdgeGraspNet [18] estimates two points on the object and uses their location and normal information to reconstruct the gripper rotation, leaving no room for our approach. It's also difficult to apply our method to works such as [17, 16] which discretize the grasp rotation and avoid direct regression. Since each hardware and detector architecture has pros and cons, in this work, we focus on investigating the benefits of our approach as an add-on, leaving the comparison against those approaches as a future work.

## 6 Conclusion

In this paper, we proposed a $SO(3)$ representation that can parameterize two planar-symmetric poses with a single parameter set by leveraging the 2D Bingham distribution. We also detailed the implementation of a neural-network-based grasp detector that uses our planar-symmetric $SO(3)$ representation. Qualitative analysis showed the benefits of our representation for a consistent network output. Finally, we performed an intensive quantitative evaluation with multiple grippers and objects in both the simulation and the real world to demonstrate efficacy of our approach.

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

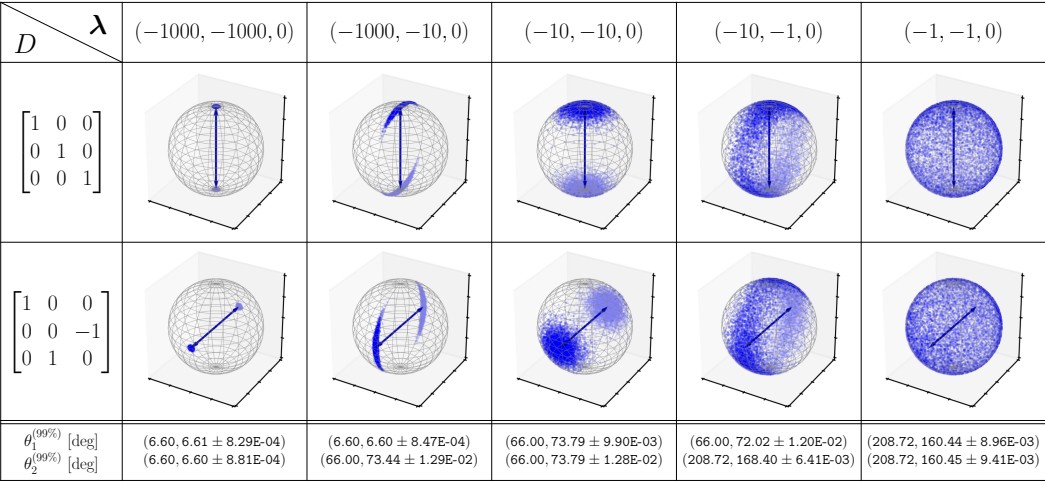

| $D$ \ $\boldsymbol{\lambda}$ | $(-1000, -1000, 0)$ | $(-1000, -10, 0)$ | $(-10, -10, 0)$ | $(-10, -1, 0)$ | $(-1, -1, 0)$ |
|---|---|---|---|---|---|
| $\begin{bmatrix} 1 & 0 & 0 \\ 0 & 1 & 0 \\ 0 & 0 & 1 \end{bmatrix}$ | | | | | |
| $\begin{bmatrix} 1 & 0 & 0 \\ 0 & 0 & -1 \\ 0 & 1 & 0 \end{bmatrix}$ | | | | | |
| $\theta_1^{(99\%)}$ [deg] 
 $\theta_2^{(99\%)}$ [deg] | $(6.60, 6.61 \pm 8.29\text{E-}04)$ 
 $(6.60, 6.60 \pm 8.81\text{E-}04)$ | $(6.60, 6.60 \pm 8.47\text{E-}04)$ 
 $(66.00, 73.44 \pm 1.29\text{E-}02)$ | $(66.00, 73.79 \pm 9.90\text{E-}03)$ 
 $(66.00, 73.79 \pm 1.28\text{E-}02)$ | $(66.00, 72.02 \pm 1.20\text{E-}02)$ 
 $(208.72, 168.40 \pm 6.41\text{E-}03)$ | $(208.72, 160.44 \pm 8.96\text{E-}03)$ 
 $(208.72, 160.45 \pm 9.41\text{E-}03)$ |

Figure 7: Visualizations of 2D Bingham distribution which is sampled by the method in Kent et al. [25]. The blue arrows represent the peak of the distribution. The parameter $D$ in Eq. 1 controls the peak direction. The $\boldsymbol{\lambda}$ parameters satisfying Eq. 5 correlate the shape of the distribution. The values of $\theta_i^{(99\%)}$ are represented by tuples whose first element is the 99th percentile approximated by Eq. 12 and second is calculated directly from 1M sampled points. The second element is in the form $\mu_{\text{MC}} \pm \sigma_{\text{MC}}$, where $\mu_{\text{MC}}$ is the mean of 100 calculation of percentiles and $\sigma_{\text{MC}}$ is the standard deviation of $\mu_{\text{MC}}$.

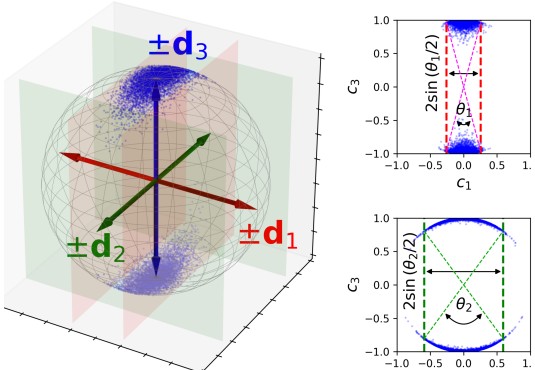

Figure 8: The visualization of the sampled points from $\mathcal{B}(\text{diag}(-50, -10, 0))$. Note that $\boldsymbol{d}_i = \boldsymbol{e}_i$ whose $i$-th component is 1 and the others are 0. In the left 3D figure, the sampled points are depicted as blue scatter points. The domain between the red planes denotes $\mathcal{A}_1(\theta_1^{(99\%)})$, which contains 99% of the sampled points. Similarly, the domain between the green planes is $\mathcal{A}_2(\theta_2^{(99\%)})$. In the right 2D figures, the upper (repr., lower) figure shows the projection of the sampled points onto the plane spanned by $(d_1, d_3)$ (repr., $(d_2, d_3)$), where the red line (repr., the green line) corresponds to the red plane (repr., the green plane) in the left figure.

# Appendices

## A Understanding of 2D Bingham Distribution

To show correlations between shape parameters, $\boldsymbol{\lambda} = (\lambda_1, \lambda_2, \lambda_3)$, and output distributions, we visualize several results in Fig. 7.

Furthermore, we conduct an intuitive analysis about shape parameters, $\boldsymbol{\lambda} = (\lambda_1, \lambda_2, \lambda_3)$. Let $A \in \text{Sym}_3$ be the parameter matrix of Bingham distribution and be diagonalized as Eq. 1. We sorted

the entries of $\boldsymbol{\lambda}$ satisfying Eq. 5. Letting $(c_1, c_2, c_3)^\top$ be the coordinates of $\boldsymbol{x}$ relative to the frame spanned by the column vectors of $D$, we get

$$(c_1, c_2, c_3)^\top = D^\top \boldsymbol{x}. \tag{9}$$

We define a subset of $\mathbb{S}^2$ as

$$\mathcal{A}_i(\theta_i) := \left\{ \boldsymbol{x} \in S^2 \mid c_i \in [-\sin(\theta_i/2), \sin(\theta_i/2)] \right\}. \tag{10}$$

The visual description of $\mathcal{A}_i(\theta_i)$ is provided in Fig. 8. Letting $X \sim \mathcal{B}(A)$ be a random variable that follows the Bingham distribution with the parameter $A$, we can define $\theta_i^{(p)} \in [0, \pi]$ as $\theta_i$ satisfying the following equation.

$$\mathrm{Prob}\left(X \in \mathcal{A}_i(\theta_i)\right) = p. \tag{11}$$

In this paper, $p = 99\%$ is used for specific calculations.

Kent [31] implies that if both $\lambda_3 - \lambda_1$ and $\lambda_3 - \lambda_2$ are sufficiently large, the distribution of $2\theta_i$ asymptotically approaches a von Mises distribution with the shape parameter $\kappa = (\lambda_3 - \lambda_i)/2$. Approximating von Mises distribution as Gaussian [32], we can yield the following approximation.

$$\theta_i^{(p)} \approx \frac{2\,\mathrm{erf}^{-1}(p)}{\sqrt{\lambda_3 - \lambda_i}} \tag{12}$$

Here $\mathrm{erf} : \mathbb{R} \to (-1, 1)$ is defined by Eq. 18, and $\mathrm{erf}^{-1} : (-1, 1) \to \mathbb{R}$ is its inverse function.

## B  2D Bingham Negative Log-Likelihood Loss

The negative log-likelihood loss [10] for the 2D Bingham distribution is defined as:

$$\mathcal{L}_{\mathrm{BNLL}}(A, \boldsymbol{v}_{\mathrm{gt}}) = -\boldsymbol{v}_{\mathrm{gt}}^\top A \boldsymbol{v}_{\mathrm{gt}} + \ln \mathcal{C}(\boldsymbol{\lambda}). \tag{13}$$

The normalizing constant $\mathcal{C}(\boldsymbol{\lambda})$ can be calculated as

$$\mathcal{C}(\boldsymbol{\lambda}) = e^c h \sqrt{\pi} \sum_{n=-N-1}^{N} w(|nh|) \, \mathcal{F}(nh, \boldsymbol{\lambda}) \, e^{nh\sqrt{-1}}, \tag{14}$$

where

$$\mathcal{F}(t, \boldsymbol{\lambda}) = \prod_{k=1}^{3} \left(-\lambda_k + t\sqrt{-1} + c\right)^{-1/2}. \tag{15}$$

The derivative $\partial \mathcal{C}/\partial \boldsymbol{\lambda}$ can be obtained by substituting $\partial \mathcal{F}/\partial \boldsymbol{\lambda}$ for $\mathcal{F}$ in Eq. 14. Let $c, h$ be defined as

$$c = \frac{N_{\min}\pi}{r^2(1+r)\omega_d}, \quad h = \sqrt{\frac{2\pi d(1+r)}{\omega_d N}}, \quad r \geq 2 \quad \text{and} \quad \frac{1}{r} \leq \omega_d \leq 1, \tag{16}$$

where $d$ is any positive number satisfying $d < c$, and $N$ is a positive integer satisfying $N \geq N_{\min}$. The function $w$ in Eq. 14 can be parametrized by $p_1$, $p_2$.

$$w(x) = \frac{1}{2}\,\mathrm{erfc}\left(\frac{x}{p_1} - p_2\right), \quad p_1 = \sqrt{\frac{Nh}{\omega_d}}, \quad p_2 = \sqrt{\frac{\omega_d Nh}{4}}, \tag{17}$$

where $\mathrm{erfc}$ is the complementary error function which defined by $\mathrm{erfc}(x) := 1 - \mathrm{erf}(x)$. Here $\mathrm{erf} : \mathbb{R} \to (-1, 1)$ is the error function defined by

$$\mathrm{erf}(x) = \frac{2}{\sqrt{\pi}} \int_0^x e^{-t^2} dt, \tag{18}$$

As described in Sato et al. [10], we set $d = c/2$, $N_{\min} = 15$, $r = 2.5$, $\omega_d = 0.5$, $N = 200$ here.

## C    Additional Training Details

Our model architecture and training pipeline follows [7]. We first created 90 meshes of daily household objects. For each object, we first sampled antipodal grasps and then randomized the grasp pose. For each grasp pose, we performed a grasp simulation in a gravity-less simulator and measured the relative pose between the hand and the object. This allows us to create a grasp pose dataset with both precision grasps and power grasps. We then project the grasp poses to the scenes, which contain multiple objects in a randomized way.

We trained the network against three labels. (i) Gravity-rejection score (trained via L2 loss) stands for how strong a grasp can resist against a gravity-direction disturbance. This label is generated in the simulator based on the approximation proposed in [7]. It serves to encourage power grasping, which is more robust than precision grasping. (ii) Grasp-validness score (trained via cross-entropy loss) stands for the probability that a grasp is a valid one. This label is generated by filling all voxels with a grasp by one and those without a grasp by zero. It serves to reject grasps that are colliding with the objects or trying to grasp parts that are too wide to grasp. Finally, (iii) per-voxel grasp rotation trained via the loss proposed in Eq. 8.

## D    Large Flat Box

The Robobiq 2F-85 gripper has a maximum grasp width of 85 mm. We, therefore, created a box object with 30 mm × 70 mm × 200 mm size. When the box is laid down, the clearance between the object and the finger surface is 7.5 mm. This corresponds to 22.6° yaw tolerance if the hand's position is perfectly aligned with the box, considering the finger's 22 mm width (See Fig. 9). The effective tolerance, however, is smaller because both the network input TSDF volume and the grasp position are discretized by the voxel size of 7.5 mm (30 cm workspace divided by 40 resolution). This makes the decluttering task challenging and requires a high position/rotation accuracy of the grasp pose.

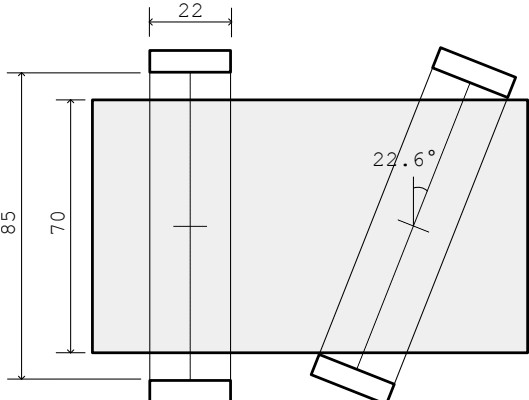

Figure 9: The combination of a 70mm-wide box and the gripper with 85 mm open width only allows a 7.5 mm translation error on each side, which is the same value as the voxel size of the network's input/output. Even when the gripper is at the perfect location, the yaw tolerance is limited to 22.6°.

## E    Additional Details of Real-Robot Evaluation

Figure 10 shows the target objects used in the real-robot evaluation described in Sec. 4.4. Among the objects, the most challenging one was 035_power_drill, with more than 1kg of weight unevenly distributed to the adversarial shape. Its slippery plastic surface also makes the grasping hard. In many cases, even when the grasp looked good, it was dropped when the robot tried to lift it. Other challenging objects include 005_tamato_soup_can and 010_potted_meat_can. While the official YCB objects are shipped with their insides vacant, we replaced them with unopened

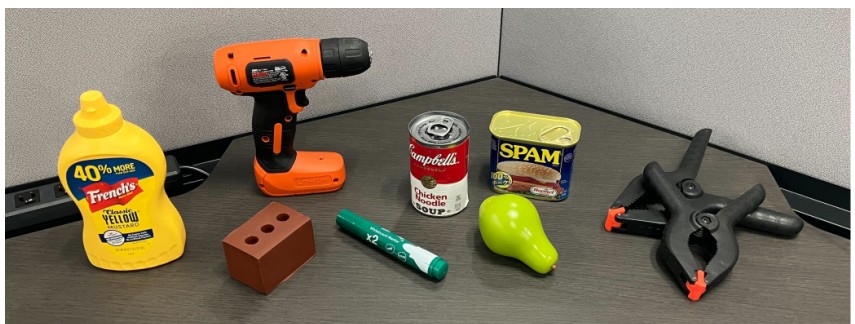

Figure 10: Target objects for real-robot evaluation.

ones to avoid damaging them. However, this makes the objects heavier (roughly 300g) and more rigid (meaning smaller contact regions). Combined with their slippy metal surface, the grasp was significantly more challenging. We also observed that `006_mustard_bottle` was challenging when it was laid down on the surface. Even though it is a popular object that is used in various evaluations, when laid down, it has a 90 mm width in the widest part and an 80 mm width in the narrowest part, leaving almost zero tolerance for our hand with an 85 mm maximum aperture. Similarly, `010_potted_meat_can` was also challenging with its 80 mm height and 95 mm width when it was laid down.

