# OpenReview forum: "A Planar-Symmetric SO(3) Representation for Learning Grasp Detection"
_robot-learning.org/CoRL/2024/Conference — CoRL 2024_

### Official Review · Reviewer_oYED · 2024-06-26

**Originality:** 3
**Technical Quality:** 3
**Clarity Of Presentation:** 3
**Potential Impact:** 3
**Recommendation:** 4
**Confidence:** 4

**Review:**

# Strengths

The idea of incorporating an antipodal symmetry (which is typical of grippers) by relaxing the representation of rotations in a probabilistic way and employing a symmetric distribution (specifically, the Bingham distribution) is original and appealing.

The experimental analysis is comprehensive, including both simulated and real-world scenarios. Multiple baselines are considered, consisting of different representations of rotations.


# Weaknesses

In my opinion, some comparisons are missing. Specifically, in the experiments performed, the two baselines considered for representing rotations are the quaternions and the rotation matrix. Both are deterministic and non-Euclidean. Instead, the proposed method is Euclidean (since the parameter space is $\mathbb{R}^9$) and probabilistic. There are other popular representations in the literature. For example, the $6$-dimensional representation from [1] is Euclidean and specifically designed for deep learning. It has even been applied in the context of grasping [2]. Another possible probabilistic baseline is using a Bingham distribution over $S^3$, where the latter is interpreted as quaternions [3]. This is similar to the proposed method, but instead of incorporating symmetries in 3D, it uses the Bingham distribution to remove the extra symmetries of the quaternion-based representation. I suggest these baselines not only for the sake of comparison, but because they would highlight whether the proposed methodology performs well due to the incorporated antipodal symmetry, or because it is a Euclidean and/or probabilistic approach, which are beneficial properties by themselves.

I find the presentation of the method (Section 3) slightly confusing, and lacking important details. For the sake of self-containment, I would have preferred the Bingham distribution being introduced in the main body, instead of the Appendix. Otherwise, most of Section 3.1 is left unexplained. The confusing point is that the expression for the Bingham distribution is anyway introduced indirectly later on, in the form of its negative log-likelihood within the loss function (Equation 3 and Equation 4). After that, some room is spent for the expression of the derivatives of the log-likelihood. Why, instead, not introducing the expression for the density of the Bingham distribution after Equation 2, and confine the expression for its derivatives to the Appendix?

[1] Zhou et al., On the Continuity of Rotation Representations in Neural Networks, CVPR 2019.

[2] Weng et al., DexDiffuser: Generating DexterousGrasps with Diffusion Models, 2024.

[3] James et al., Bingham Policy Parameterization for 3D Rotations in Reinforcement Learning, 2022.

# Additional Comments
Typo at line 72: space missing before ‘(BMM)’.

I suggest reducing the length of the caption of Figure 1. Right now, the abstract appears more than half-way through page 1 because of this caption, which in my opinion is a bad typographical practice.

**Quality Of The Limitations Section:**

3

**Questions For Rebuttal:**

I would like the authors to comment on the weakness raised above regarding the comparison with other representations for rotations.

**Robotics Focus:**

4

**Summary Of Paper:**

The work proposes to deploy a 2D Bingham distribution in order to incorporate an antipodal planar symmetry in the context of grasping.

**Summary Of Recommendation:**

I believe that despite the weaknesses mentioned above, the work presents an original and interesting idea. Therefore, I lean towards accepting it for publication.

---

### Official Review · Reviewer_pqFb · 2024-07-19
**Interesting idea, but empirical results not compelling enough**

**Originality:** 3
**Technical Quality:** 2
**Clarity Of Presentation:** 4
**Potential Impact:** 2
**Recommendation:** 2
**Confidence:** 5

**Review:**

Strength:
The paper studies the bimanual symmetry of the gripper, a prominent yet often overlooked aspect of symmetry in robotic manipulation. The proposed representation is general and can potentially be applied in many different learning frameworks.

Weakness:
Utilizing the bilateral symmetry of the gripper is not a new idea, although the paper proposes a novel method for doing so. It has been explored through various methods such as constraining the symmetry in the loss function [4], flipping the contact normal [17], using the quotient representation [A], or performing data augmentation.

Moreover, the reported grasping success rate in Table 1 is significantly lower than existing SOTA grasping methods like GIGA [5] and Edge Grasp Network [17]. It is important for the proposed method to either boost the performance of one of the SOTA grasping methods or improve an existing method’s performance to reach a similar level as the SOTA baselines.

Although the simulation environments are not the same comparing this paper and the GIGA benchmark used in [5, 17], the object sets are the same, i.e., both use the VGN dataset. I understand that it is not a direct comparison to compare Table 1 in this paper and the results in [5, 17], so it would be helpful to run [5, 17] in this paper's environment or run the proposed method in the GIGA benchmark.

[A] Zhu, Xupeng, et al. "Sample efficient grasp learning using equivariant models." RSS (2022).

**Quality Of The Limitations Section:**

3

**Questions For Rebuttal:**

Questions:
1. While the paper compares the learning of bilateral symmetry using a loss term as proposed in [4], it would be beneficial to also discuss and compare other methods that leverage bimanual symmetry.
2. Given the focus on grasping, my biggest concern is the performance shown in Table 1, which underperforms some existing baselines, as mentioned in the weaknesses section. Can the authors comment on this? Is the experimental environment more challenging than those used in other works? Can the authors show the performance of the proposed method in one of the standard benchmarks, like the GIGA benchmark [5]?
3. Similarly, in the real world, the proposed method only achieves a 70% grasp success rate. What are the main failure modes?
4. It would be helpful to include a figure in the appendix to show the selected 9 objects in the real-world experiment.
5. Can the authors comment on why the proposed method performs better than the rotation matrix in the cases shown in Figure 6?
6. In the limitations section, the authors claim, "Another limitation is that our work applies only to grasp detectors that explicitly regress the gripper rotation." I wonder if one can use a sample-based method and send the Bingham distribution as the input to the network? This might also solve the multimodality problem of the grasp rotation distribution, which is something the regression-based method will suffer from.

**Robotics Focus:**

4

**Summary Of Paper:**

This paper proposes an SO(3) representation using a 2D Bingham distribution for grasp detection, which captures the bilateral symmetry of a parallel jaw gripper. By comparing it against other rotation representations (i.e., rotation matrix and quaternion), the paper demonstrates that the proposed representation achieves higher grasp success rates than these traditional representations.

**Summary Of Recommendation:**

While the proposed method is intriguing, the empirical results currently presented are not sufficiently compelling. If the bottleneck is the backbone network, I would suggest the authors apply their method in one of the SOTA grasp detection algorithms and improve the experimental results.

---

### Official Review · Reviewer_Ueu5 · 2024-07-21
**Focused on a single contribution but clear**

**Originality:** 3
**Technical Quality:** 4
**Clarity Of Presentation:** 4
**Potential Impact:** 3
**Recommendation:** 3
**Confidence:** 4

**Review:**

Strength:
The paper focused on a particular contribution, which is the modeling of the antipodal symmetry in order to resolve the inconsistency of the learned representations when the grasping pose is modeled with an SO(3) rotation. The methodology is clear, and the experimental results validated the effectiveness of the proposed representation. The analysis in section 4.1 helps readers understand the uncertainty and the distribution of the learned grasp poses with and without the proposed representation, which highlights the value of this paper.

Weakness:
The framework is mostly built upon an existing work ([7] in paper). Since the contribution of this paper is mostly about rotation modeling, the context of the other part of the framework and the experimental setup are brief, which could make it harder for the readers to understand the whole setup. The paper could be made more self-contained. For example, the gravity-rejection score is briefly mentioned, but the motivation and usefulness of the score are not explained. For the experiment, it is also not clear what data and how much data is used for the training.

**Quality Of The Limitations Section:**

3

**Questions For Rebuttal:**

I did not understand the experiment setup in section 4.4. Could you explain this paragraph further? "We then iterated over each output voxel in the gravity-rejection score order to find the first applicable robot trajectory that fulfills: (i) the gripper linearly in the z direction 30 cm to the pre-grasp pose, (ii) the gripper moves forward for 10 cm to grasp the object, (iii) the gripper is linearly lifted by 30 cm." What do you mean by  "iterating over each output voxel in the gravity-rejection score order to find the first applicable robot trajectory"?

**Robotics Focus:**

4

**Summary Of Paper:**

This paper proposed to model the grasping detection with a planar-symmetric SO(3) representation, parameterized using 2D Bingham distribution. In this way, it can handle the 180-degree ambiguity of the symmetric grippers and avoid inconsistencies in the learned representations caused by this symmetry, while being able to model the uncertainty of the estimated grasping pose.

**Summary Of Recommendation:**

The contribution is focused and clear, though relatively simple conceptually. The proposed representation could be valuable in general grasping pose modeling.

---

### Decision · Program_Chairs · 2024-09-04

**Decision:**

Accept

**Comment:**

Here are the strengths and weaknesses of the paper as identified by the reviewer and area chair:

Strength:
- the direction of leveraging the symmetry in robotic grippers for better grasp detection is interesting and relevant
- The approach of how to incorporate this symmetry is novel
- the paper contains experiments in simulation and the real world

Weaknesses:
- while there are substantial experiments reported, the choice of baselines should be improved. For example, there are other ways to represent rotations. Or there are other SOTA grasp detectors that could be evaluated on the proposed dataset to understand their performance relative to the proposed approach.
- The dataset on which the authors compared is not one of the standard dataset and therefore prevents the reviewers to understand how much the proposed approach improves on existing grasp synthesis methods for parallel yaw grippers. It would be good if the authors could run their approach on existing datasets while at the same time also providing more details on the dataset they evaluated on.
- the reviewers make several suggestions to improve writing clarity of the paper

For the rebuttal, the authors have clarified their contributions and baselines. Specifically, they compare against and significantly outperform [7]  (rotation matrices - IROS paper) in both sim and real. The authors also run a set of (approximate) comparisons against [17,4]. However, this analysis is not quite at the level of rigor as the comparison to [7] (in my mind especially a comparison to [17] seems important) to demonstrate that they actually have better or at least comparable grasp performance given the new representation. A watertight comparison against [17,4] would need to be done following the same methodology as the comparison against [7].

For the camera-ready version I recommend the authors to make it more clear that the comparison against using rotation matrices is actually from a different paper and therefore a SOTA method. The authors should only include any conclusion on the performance compared to [17,4] if they follow the exact same methodology and data as done for [7]. I don't recommend to include the arguments laid out in the rebuttal.